# Changing Spectrum of Opportunistic Illnesses among HIV-Infected Taiwanese Patients in Response to a 10-Year National Anti-TB Programme

**DOI:** 10.3390/jcm8020163

**Published:** 2019-02-01

**Authors:** Chun-Yuan Lee, Pei-Hua Wu, Po-Liang Lu, Hung-Chin Tsai

**Affiliations:** 1Division of Infectious Diseases, Department of Internal Medicine, Kaohsiung Medical University Hospital, Kaohsiung Medical University, Kaohsiung 80756, Taiwan; leecy8801131@gmail.com (C.-Y.L.); hua830820@gmail.com (P.-H.W.); 2Graduate Institute of Medicine, Kaohsiung Medical University, Kaohsiung 807, Taiwan; 3Center for Infectious Disease and Cancer Research (CICAR), Kaohsiung Medical University, Kaohsiung 807, Taiwan; 4School of Medicine, College of Medicine, Kaohsiung Medical University, Kaohsiung 807, Taiwan; 5Department of Biological Science and Technology, College of Biological Science and Technology, National Chiao Tung University, Hsin Chu 30010, Taiwan; 6Department of Laboratory Medicine, Kaohsiung Medical University Hospital, Kaohsiung 80756, Taiwan; 7Division of Infectious Diseases, Department of Medicine, Kaohsiung Veterans General Hospital, Kaohsiung 81362, Taiwan; 8Faculty of Medicine, School of Medicine, National Yang-Ming University, Taipei 11221, Taiwan; 9Department of Parasitology, Kaohsiung Medical University, Kaohsiung 807, Taiwan

**Keywords:** AIDS-related opportunistic infections, HIV, Mycobacterium, tuberculosis, CD4 lymphocyte count

## Abstract

The current trends and spectrum of acquired immunodeficiency syndrome (AIDS)-related opportunistic illnesses (AOIs) among newly diagnosed human immunodeficiency virus (HIV)-infected patients after the implementation of the 2006–2015 national anti-tuberculosis (TB) programmes in Taiwan remain unknown. We retrospectively reviewed 1757 patients at two centres in southern Taiwan between 2001 and 2015. Based on the anti-TB programme, patients were classified into periods 1 (2001–2005), 2 (2006–2010), and 3 (2011–2015). We further analysed factors associated with *Mycobacterium tuberculosis* (MTB) at presentation and during follow-up. The overall AOI incidence rate (23.6%) remained unchanged across the periods, with 81.4% of AOIs occurring at presentation. *Pneumocystis jirovecii* pneumonia was the leading AOI across the periods. MTB declined significantly from period 1 to period 3 (39.3% vs. 9.3%). Age and CD4+ cell count <200 cells/µL (vs. ≥501) were the risk factors associated with MTB at presentation, whereas period 2/3 (vs. period 1) was the protective factor. Intravenous drug use (vs. homosexual contact) was the risk factor associated with MTB during follow-up, and period 3 (vs. period 1) was the protective factor. AOI statistics in Taiwan must be closely monitored for fluctuations. Although MTB decreased substantially after implementation of the anti-TB programmes, additional efforts to reduce MTB are required.

## 1. Introduction

The development of antiretroviral therapy (ART) has substantially decreased HIV-associated morbidity and mortality [1,2]. However, AIDS-related opportunistic illnesses (AOIs) are common at HIV presentation or during follow-up in both developing [3,4] and developed [5,6,7] countries. Moreover, AOIs continue to be a leading cause of hospitalisation [8,9] and mortality [10,11] among HIV-infected patients.

AOIs are infections or malignancies that occur more frequently and are more severe in hosts with weakened immune systems by HIV, such as *Pneumocystis jirovecii* pneumonia (PJP), toxoplasmosis of brain, and Kaposi’s sarcoma [12]. AOIs vary widely depending on the studied region (such as *Mycobacterium tuberculosis* (MTB) as the most common aetiology in Philippines [13], India [14], and China [8]; PJP in Japan [13] and Australia [15]; esophageal candidiasis in Poland [16]; and cerebral toxoplasmosis in Lebanon [17]), the environmental prevalence of the microorganisms (such as MTB [18,19,20,21], *Penicillium marneffei* [8,22], *Toxoplasma gondii* [23], and *Cryptococcus neoformans* [24,25]), and the definitions of AOI [19,26,27,28,29]. The AOI spectrum may also change with improved survival due to ART. For example, ART may result in a specific opportunistic illness at high CD4+ cell counts [1,28,30]. Therefore, continuously monitoring the incidence and spectrum of AOIs is vital in all geographical areas.

Tuberculosis (TB) has had a major public health impact on southeast Asian countries [31,32], including Taiwan [33,34]. In line with the Global Plan to Stop TB 2006–2015, advocated by the World Health Organisation (WHO), Taiwan launched a programme aimed at halving TB incidence from 72.5 to 36 cases per 100,000 people by 2015, known as the National Mobilisation Plan to Halve TB in 10 Years (Phase 1, 2006–2010 [35]; Phase 2, 2011–2015 [36]). The Phase 1 5-year programme was approved by Executive Yuan in 2006. With the implementation of the Phase 1 programme, the number of new cases of MTB in Taiwan each year has declined from 16,472 in 2005 to 14,265 in 2008 [36]. After assessing the results of the Phase 1 programme, in coordination with The Global Plan to Stop TB 2006-2015 recommended by the Stop TB Partnership, the Department of Health approved the Phase 2 5-year programme to reduce the TB burden in Taiwan. The 10-year plan aimed to develop a comprehensive national programme that strengthened resources at all levels of TB control and prevention, to intensify case-finding and reporting strategy, to improve the quality of diagnosis and therapy of tuberculosis, to implement case management, and to scale up the knowledge and awareness of the pubic to tuberculosis prevention and control. Consequently, the TB incidence in the general population declined from 72.5 cases per 100,000 people in 2005 [37] to 45.7 cases per 100,000 people in 2015 [38], representing a 37.0% reduction from the 2005 incidence level. 

However, adequate data to investigate the evolving AOI incidence rate and spectrum among HIV-infected patients in Taiwan after the implementation of the 2006–2015 national TB programmes are lacking, particularly with regard to MTB disease [39,40,41,42]. Moreover, a study conducted in southern Africa revealed a possible interaction between MTB and other AOIs, such as cryptococcal infection and PJP, in HIV-infected patients [18], further reinforcing the need to update AOI surveillance in Taiwan following the 10-year national TB programme. In addition, previous studies conducted in Western Europe found advanced HIV infection, non-white ethnicity, migrants from high TB incidence countries, and intravenous drug user (IVDU) status as risk factors of MTB disease in HIV-infected patients [43,44,45]. However, these studies may not be applied to Taiwan, which has a higher TB incidence rate than Western Europe [31].

Therefore, the purpose of the present study was to estimate the evolving AOI incidence rate and spectrum among newly diagnosed cases of HIV infection at the two largest referral centres for patients with HIV in southern Taiwan, in response to a 10-year national anti-TB programme. Patients were stratified into three periods: Period 1 (before programme, 2001–2005), period 2 (Phase 1 programme, 2006–2010), and period 3 (Phase 2 programme, 2011–2015). We further analysed factors associated with MTB disease at presentation and during follow-up.

## 2. Materials and Methods

### 2.1. Design, Setting, and Data Source

This retrospective cohort study was conducted at Kaohsiung Medical University Hospital (KMUH) and Kaohsiung Veterans General Hospital (VGHKS), the two largest referral centres for patients with HIV in southern Taiwan. The medical records of all eligible patients were reviewed by trained staff, who retrieved each patient’s age, sex, date of first HIV-related medical visit, risk factors for HIV-1 infection, serum CD4+ cell counts, HIV viral load (VL) at presentation, HIV stage at presentation, and AOIs with their associated outcomes.

### 2.2. Participants

First, we screened all first-time patients seeking HIV-related care in person at VGHKS and KMUH from January 1, 2001 to December 31, 2015. Those who had received follow-up care at both hospitals were counted only once. We excluded patients without a new diagnosis, those aged <15 years, and those seen only once as outpatients (and thus lost to follow-up) during the observation period. Next, the enrolled patients were classified into three groups according to the dates of their initial HIV-related visit: Periods 1 (before programme; January 1, 2001 to December 31, 2005), 2 (Phase 1 programme; January 1, 2006 to December 31, 2010), and 3 (Phase 2 programme; January 1, 2011 to December 31, 2015). The demographic and laboratory variables, comorbidities, AOI events, and associated outcomes of the patients were compared in each study period. Finally, factors associated with MTB disease at presentation and during follow-up were analysed. This study was approved by the VGHKS and KMUH ethics committees (VGHKS17-CT4-20, KMUHIRB-SV(II)-20170003) and adhered to the ethical principles of the Declaration of Helsinki. The informed consent requirement was waived by the institutional review boards of both participating hospitals.

### 2.3. Definitions

There is no consensus on the definition of a patient newly diagnosed with HIV infection [46]. In this study, patients considered eligible for inclusion were those who had received their initial diagnosis at a participating hospital or who had been diagnosed elsewhere but were referred to one of these hospitals within 1 month of diagnosis and for whom medical records were available [47].

The date of enrolment was defined as the date of the first visit on which the patient sought HIV-related care at either of the participating hospitals. The date of an AOI event was defined as the date of the first visit on which the patient sought care for an AOI. The final patient observation time was 6 months from the end of each study period, at death, or at the final outpatient visit or hospitalisation for patients who were lost to follow-up (whichever occurred first).

The CD4+ cell count, HIV VL, and baseline laboratory test values from within 6 months of enrolment and those closest to the date of HIV diagnosis were used [5]. The HIV infection stage at presentation was recorded as a number between 0 and 3, according to the 2014 case definition for HIV infection from the US Centers for Disease Control and Prevention (CDC) [48].

The MTB incidence at presentation was defined as the number of patients who developed MTB at ≤90 days of HIV diagnosis [49]. The MTB incidence during follow-up was defined as the number of patients who did not develop MTB disease at ≤90 days of HIV diagnosis, but developed it during the follow-up period.

### 2.4. AOI Diagnosis

AOIs were defined according to the CDC’s 1993 AIDS case definition [12], with the addition of *Talaromyces marneffei* infection, which is a common AOI in southeast Asia, southern China, and Taiwan [8,22]. Because definitive diagnoses of specific AOIs, such as *Toxoplasma* encephalitis and progressive multifocal leukoencephalopathy (PML), are difficult, a presumptive diagnosis was generally made in this study. *Toxoplasma* encephalitis was empirically diagnosed on the basis of clinical and radiographic responses to specific anti-*Toxoplasma gondii* therapy, in the absence of a likely alternative diagnosis. PML was empirically diagnosed on the basis of compatible clinical manifestations and brain magnetic resonance imaging findings, also in the absence of a likely alternative diagnosis. Cytomegalovirus (CMV) infection in the gastrointestinal tract was diagnosed on the basis of compatible histological findings, and CMV retinitis was diagnosed on the basis of compatible retinal changes observed in an ophthalmoscopic examination, with positive CMV DNA detected by polymerase chain reaction (PCR) in vitreous or aqueous humour specimens. CMV pneumonitis was diagnosed on the basis of compatible clinical and radiological findings, and identification of CMV inclusion bodies in lung tissue or cytological specimens, and CMV neurological disease was confirmed on the basis of compatible clinical syndromes and positive CMV DNA detected in cerebrospinal fluid by PCR. A definitive diagnosis of PJP requires the detection of organisms in the respiratory tract (lung tissue, bronchoalveolar lavage fluid, or induced sputum) samples by histopathology, cytopathology, or PCR. Here, however, a presumptive diagnosis of PJP was made if patients presented with compatible clinical manifestations and computed tomography findings of patchy ground-glass attenuation, and clinically responded to anti-PJP treatment. TB disease was diagnosed on the basis of a positive MTB culture, or clinical and radiological signs that were consistent with TB and improvement with standard anti-TB therapy. Other opportunistic infections and malignancies were diagnosed according to the criteria recommended in the CDC’s 1993 AIDS case definition [12]. If patients exhibited more than one AOI episode during the same outpatient visit or hospitalisation, the episodes were recorded as a single AOI event. AOI episodes occurring during different outpatient visits or hospitalisations, with no symptoms at the immediately preceding outpatient visit or hospitalisation, were registered as separate events, and the associated laboratory results and outcomes were recorded sequentially.

### 2.5. Outcomes of Interest

The primary outcomes of interest were the AOI incidence rate and spectrum in the three study periods. The secondary outcomes were factors associated with MTB disease at presentation and during follow-up. We also compared the demographics and clinical manifestations of MTB disease in the three study periods.

### 2.6. Statistical Analysis

Categorical variables in the three groups were compared using the χ^2^ or Fisher’s exact test, whereas continuous variables were compared by variance analysis.

In the univariable analysis, the CD4+ cell count at presentation, rather than the HIV stage at presentation, was used because of the considerable collinearity between these two variables. Univariable analysis and multivariable logistic regression were employed to identify factors associated with MTB disease at presentation. All variables used in the univariable analysis were selected for subsequent multivariable logistic regression.

The probability of MTB-free survival, stratified by the three study periods, was estimated using Kaplan–Meier survival curves and log-rank tests. Cox regression analysis was employed to identify the correlates of MTB disease during follow-up. All variables identified in the univariable analysis were included in the multivariable analysis.

Odds ratios (ORs) and hazard ratios (HRs) with their respective 95% CIs were used to estimate each variable’s effects and the directions of associations. All tests were two-tailed, and *p* < 0.05 was considered significant. Statistical analysis was performed using SPSS (version 22.0; IBM Corp., Armonk, NY, USA).

## 3. Results

### 3.1. Patient Characteristics

Of the 2956 screened patients, 1199 were excluded because they were not newly diagnosed (*n* = 1077), not followed-up a second time (*n* = 120), or aged <15 years (*n* = 2). Of the remaining 1757 enrolled patients, 194 (11.0%), 492 (28.0%), and 1071 (61.0%) were included in periods 1, 2, and 3, respectively, as shown in Figure 1.

The baseline demographic characteristics and comorbidities of the three groups are summarised in Table 1. The median (interquartile range (IQR)) duration of observation was 741 (961) days. 97.3% of the patients were male. The mean (standard deviation (SD)) age at presentation was 30 (10) years. The mean age declined significantly from 32 (10) years in period 1 to 31 (10) and 29 (9) years in periods 2 and 3, respectively (*p* < 0.001). The main mode of transmission was homosexual contact, which increased significantly from 67% in period 1 to 82.1% in period 3 (*p* < 0.001). The mean (SD) CD4+ cell count at presentation was 283 (214) cells/L. The mean (SD) CD4+ cell count significantly declined from 319 (229) cells/L in period 1 to 275 (204) cells/L in period 3 (*p* = 0.031). The overall incidence of AIDS at presentation was 38.3%, with no significant differences among the three periods.

In summary, the following significant trends were observed over the three successive study periods: A decrease in patient age, an increase in homosexual contact, and a decrease in the CD4+ cell count at presentation. AIDS at presentation remained unchanged throughout the three study periods.

### 3.2. AOI Incidence

In total, 635 AOI episodes involving 460 AOI events occurred, for an average of 1.38 episodes per AOI event over 3975 person-years of observation. The overall proportion of patients who developed AOIs during the study period was 23.6% (414/1757). This did not decline significantly, showing decreases from 25.8% in period 1 and 26.2% in period 2 to 21.9% in period 3 (*p* = 0.134). In addition, the majority of AOIs (337/414, 81.4%) occurred at presentation. The overall prevalence of AOIs at presentation was 19.2%, which did not decline significantly, showing a decrease from 21.6% in period 1 to 18.2% in period 3 (*p* = 0.4; Table 1). In summary, the results suggest that the AOI incidence remained unchanged across the three study periods.

### 3.3. AOI Spectrum

The spectrum of 635 AOI episodes from 460 events over the three study periods is presented in Table 2. Overall, the five most common AOIs during the study period (2001–2015) were PJP (57.8%), MTB disease (17.0%), cytomegalovirus (CMV) diseases (12.0%), candidiasis (11.7%), and wasting syndrome (10.2%). In total, 35 (7.6%) opportunistic malignancies were found (Kaposi sarcoma, *n* = 16; lymphoma, *n* = 18; and invasive cervical cancer, *n* = 1). Although PJP was the leading cause of AOIs in all three study periods (58.9%, 57.1%, and 58.0% in periods 1, 2, and 3, respectively), the incidence of other specific AOIs changed significantly. The incidence of AOIs with MTB disease declined significantly, from 39.3% in period 1 to 21.8% and 9.3% in periods 2 and 3, respectively (*p* < 0.001). The incidence of AOIs with candidiasis also declined significantly, from 23.2% in period 1 to 9.3% in period 3 (*p* < 0.014). Although not statistically significant, there was a notable increase in the incidence of CMV disease from 5.4% in period 1 to 14.8% in period 3 (*p* = 0.078). In summary, the specific AOI incidence changed significantly across the three study periods for candidiasis and MTB only.

### 3.4. Factors Associated with MTB Disease at Presentation

Table 3 lists the factors associated with MTB disease at presentation. Logistic regression analysis revealed that the following risk factors were associated with MTB disease at presentation: Age (per 10-year increase; adjusted odds ratio (AOR), 1.40; 95% confidence interval (CI), 1.08–1.79, *p* = 0.009) and CD4+ cell count at presentation (<200 vs. ≥501 cells/µL; AOR, 7.71; 95% CI, 2.20–27.05, *p* = 0.001). Period 2 (vs. period 1; AOR, 0.43; 95% CI, 0.22–0.83, *p* = 0.012) and period 3 (vs. period 1; AOR, 0.16; 95% CI, 0.08–0.32, *p* < 0.001) were factors protecting against MTB disease at presentation.

### 3.5. Factors Associated with MTB Disease During Follow-Up

MTB-free survival was significantly higher in patients who were diagnosed in period 3 than patients who were diagnosed in periods 1 and 2 (*p* < 0.001; Figure 2). Table 4 lists the factors associated with MTB disease during follow-up. Cox regression analysis revealed intravenous drug use (vs. homosexual contact; adjusted hazard ratio (AHR), 5.63; 95% CI, 1.44–21.99, *p* = 0.013) as a risk factor and period 3 (vs. period 1; AHR, 0.18; 95% CI, 0.04–0.91, *p* = 0.038) as a protective factor associated with MTB disease during follow-up.

### 3.6. Characteristics of 78 MTB Disease Cases

Baseline demographic characteristics, laboratory data, and extrapulmonary manifestations of MTB disease in the three study periods are summarised in Table 5. In total, 78 MTB disease events were recorded (22, 32, and 24 in periods 1, 2, and 3, respectively; Table 5). The mean (SD) age at event was 36.1 (10.2) years. 93.6% of these patients were male. Overall, 61 patients (78.2%) presented at an advanced disease stage (CD4+ cell count < 200 cells/L), 66 patients (84.6%) developed the disease within 90 days of enrolment, and 31 patients (39.7%) had extrapulmonary TB. The 78 patients did not differ significantly in terms of demographic characteristics or disease manifestations.

## 4. Discussion

In our retrospective cohort study of 1757 patients who were newly diagnosed with HIV, the overall AOI incidence rate (23.6%) remained unchanged across the three study periods. Notably, the majority (81.4%) of AOIs occurred at presentation, thus emphasising the persistent problem of late HIV presentation in Taiwan (Table 1). The findings pertaining to the changing spectrum of AOIs in this study (Table 2) contrast those of a study conducted in northern Taiwan from 1994 to 2004, which indicated that the AOI spectrum remained unchanged [39]. This disparity may stem from the difference in study period, given that the National Mobilisation Plan to Halve TB in 10 Years had not been implemented when the other study was conducted [39]. These findings reemphasise the importance of closely monitoring the incidence rate and spectrum of AOIs to assist primary physicians in staying informed regarding the latest local AOI trends.

Consistent with our findings, PJP was also the most common AOI in Japan [13]. However, regional differences in the AOI spectrum have been reported [8,13,14,15,16,17]. The observed regional differences in AOI distribution are likely attributable to a complex interaction between socioeconomic setting [50], geographical differences [8,51], genetic susceptibility [15], the implementation of highly active ART (HAART) [39], the prevalence of specific microorganisms [8,18,19,20,21,22,23,24,25], different definitions of AOIs [19,26,27,28,29], and different public health intervention methods, as also reported here. In contrast to other studies conducted in northern Taiwan, noting candidiasis as the most common AOI aetiology, followed by PJP [39,40], this study found that PJP was the leading aetiology. This discrepancy may reflect the use of different definitions of candidiasis as the cause of AIDS. In this study, only patients with candidiasis of the oesophagus and respiratory tract were enrolled. In the other two studies, however, patients with oropharyngeal candidiasis (oral thrush) were also enrolled, which may have increased the AOI incidence attributable to candidiasis [39,40]. However, according to the CDC’s 1993 AIDS case definition, oral thrush should be classified as category B, which includes diseases caused by a cell-mediated immunity defect rather than by AIDS [12]. Therefore, the present study provides novel information regarding the AOI spectrum during the HAART era in Taiwan. The declining incidence of candidiasis as an AOI may also reflect an increasing recourse to empirically prescribed fluconazole by clinicians for HIV-infected patients presenting with oral thrush, without undertaking endoscopic examination for oesophageal candidiasis. This may have caused the incidence of candidiasis as an AOI to be underestimated in the present study.

We observed a significant reduction in MTB disease as the cause of AOIs over the successive study periods. In addition to the National Mobilisation Plan to Halve TB in 10 Years, the implementation of which began in 2006, this reduction may be attributable to several other policy interventions, including voluntary HIV counselling and testing for early HIV diagnosis beginning in 1997, updated recommendations to initiate HAART at higher CD4+ cell count thresholds (<200 cells/mm^3^ in 2006, <350 cells/mm^3^ in 2010, and <500 cells/mm^3^ in 2013), and the implementation of an HIV case management programme in 2007, which provided health consultation, psycho-social support, and general client education to help link HIV-infected patients to medical care.

However, several possibilities exist to further reduce the incidence of MTB disease in HIV-infected individuals. First, we observed that 84.6% of MTB disease developed within 90 days of HIV diagnosis and that advanced HIV infection was a risk factor associated with MTB disease at presentation (Table 3), consistent with the findings of a study in Nigeria [21]. Late HIV presentation remains a persistent problem in Taiwan [7]. Therefore, the first priority is to diagnose HIV–MTB-coinfected patients early, which requires the collaborative TB/HIV activities recommended in WHO guidelines [52], focusing particularly on early HIV diagnosis and intensified TB case investigations to further decrease the TB burden on HIV-infected patients in Taiwan [53].

To control MTB disease during follow-up, IVDU populations should receive special emphasis (Table 4). The finding that IVDUs are at risk of MTB disease is consistent with the findings of a nationwide survey conducted in the United Kingdom [45]. Alcoholism, homelessness, and difficulties in accessing care are possible explanations [45,54]. In Taiwan, however, the higher rate of MTB disease among IVDUs may be aggravated by the policy of targeted HIV-related intervention for prisoners, which includes active HIV surveillance and the centralised incarceration of HIV-infected prisoners. Considering the high rate of incarceration history among IVDUs, the interplay of factors, such as overcrowding, poor ventilation, and HIV infection, can contribute to the circulation of certain MTB strains among HIV-infected prisoners [55,56]. Further investigation of circulating strains and transmission dynamics inside prisons is urgently required in Taiwan, along with public strategies to block MTB transmission in crowded prisons and implement preventive isoniazid therapy for latent tuberculosis infection [57] among HIV-infected IVDUs.

Extrapulmonary TB has been reported to be more common among HIV-infected patients than among uninfected patients (odds ratio (OR): 1.3; 95% CI, 1.05–1.6) [58], with extrapulmonary TB declining in inverse proportion to CD4+ cell counts [59]. The patients with HIV–MTB coinfection in this study exhibited a lower extrapulmonary TB rate than the patients in the studies conducted in the pre-HAART era in Taiwan [41,42] (previous study vs. present study; 75.8–83.3% vs. 39.7%), potentially because CD4+ cell counts at MTB disease diagnosis were higher in this study than in other studies (CD4+ cell counts in the present study vs. previous study; 78 vs. 21–37 cells/L) [30,31]. However, the relative proportion of extrapulmonary TB remained unchanged throughout the three study periods (Table 5). This indicates that although the incidence of MTB disease in HIV-infected patients decreased significantly across the study periods, the majority of MTB events occurred when patients were severely immunocompromised (78.2% cases at CD4+ cell count < 200 cells/L), which could contribute to an increased rate of extrapulmonary TB and perhaps delay MTB disease diagnosis in patients with HIV [59,60]. Therefore, primary physicians should consider the protean manifestations of extrapulmonary TB during the treatment of HIV-infected patients in areas with a high prevalence of TB, despite the declining burden of AOIs with MTB disease in Taiwan [37,38].

The major strength of this study is its evaluation of long-term trends in the AOI incidence rate and spectrum in a large cohort of Taiwanese patients over a span of 15 years. To our knowledge, this is also the first study to identify intravenous drug use as a risk factor for MTB disease during follow-up among HIV-infected individuals in Taiwan. However, our research has several limitations. First, some important information could not be retrieved from retrospective chart review, such as contact history of TB index cases, whose behavioural and clinical characteristics have been reported to be associated with infectiousness of *Mycobacterium tuberculosis* [61]. Second, although the TB incidence rate have declined throughout Taiwan, southern Taiwan still has a higher TB incidence rate than northern Taiwan [34,62]. Therefore, the results of the present study conducted in southern Taiwan may not be generalisable to all of Taiwan. Third, the diagnosis of certain AOIs, such as oesophageal candidiasis and CMV infection of the gastrointestinal tract, may have been underestimated because endoscopy was not performed for critically ill or intubated HIV-infected patients. Finally, the median (IQR) observation period of the present study was 741 (961) days, which may have been too short and may have underestimated the incidence of specific AOIs associated with less severe immunosuppression, particularly in the post-HAART era [30].

## 5. Conclusions

This study indicated that the overall AOI incidence remains a persistent problem in Taiwan, particularly at the presentation of HIV infection. However, the spectrum of AOIs fluctuated during the observation period, reinforcing the need to closely monitor evolving trends. Although the MTB disease incidence decreased substantially after implementation of the National Mobilisation Plan to Halve TB in 10 Years, additional efforts to reduce MTB disease at presentation and during follow-up are required. This includes intensified HIV diagnostics at earlier stages through active surveillance of HIV to avoid progression to AIDS, public intervention to block transmission of MTB strains possibly circulating in crowded prisons, and preventive isoniazid therapy for latent TB infection among IVDUs. Furthermore, physicians should remain alert to early manifestations of extrapulmonary TB in severely immunosuppressed HIV-infected patients.

## Figures and Tables

**Figure 1 jcm-08-00163-f001:**
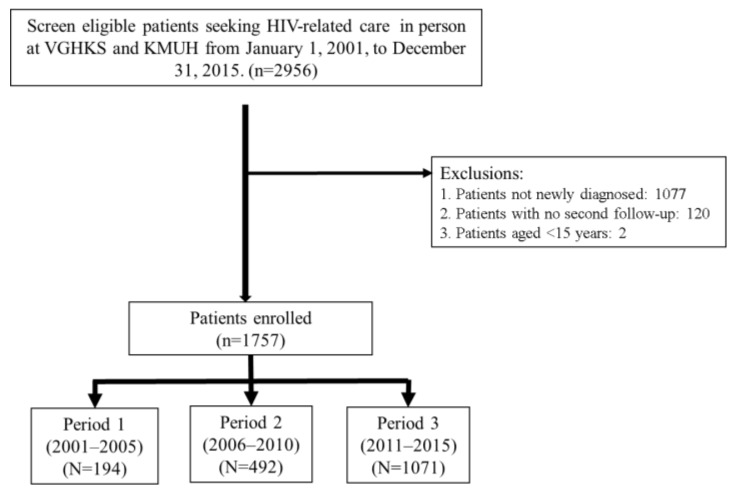
Schematic of the study population.

**Figure 2 jcm-08-00163-f002:**
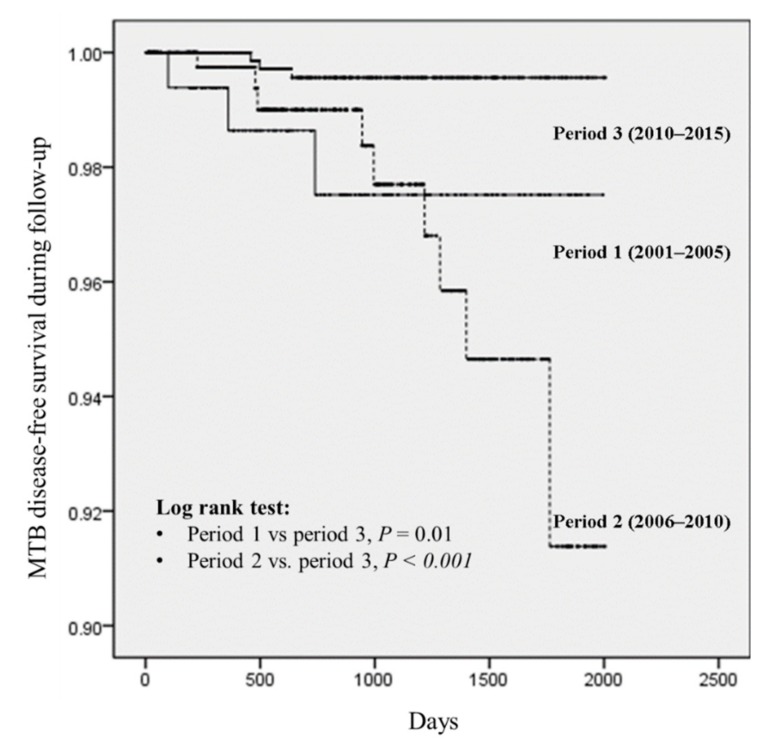
Cumulative *Mycobacterium tuberculosis* (MTB)-free survival among patients newly diagnosed with HIV, stratified according to the three study periods. The curves demonstrate poorer MTB-free survival among patients in periods 2 and 1 compared with those in period 3 (log-rank test, *p* = 0.01 and *p* < 0.001, respectively).

**Table 1 jcm-08-00163-t001:** Demographic characteristics and HIV-related variables for 1757 patients newly diagnosed with HIV infection from 2001 to 2015.

	Total (2001–2015) *n* = 1757	Period 1 (2001–2005) *n* = 194	Period 2 (2006–2010) *n* = 492	Period 3 (2011–2015) *n* = 1071	*p*-Value
Follow-up period, median (IQR; days)	741 (961)	724 (880)	551 (893)	829 (981)	<0.001
Male, *n* (%)	1709 (97.3)	186 (95.9)	468 (95.1)	1055 (98.5)	<0.001
Mean age at presentation of HIV, years (SD)	30.21 (9.75)	31.54 (10.09)	31.43 (10.32)	29.41 (9.33)	<0.001
HIV transmission route, *n* (%)					
Homosexual	1367 (77.8)	130 (67.0)	358 (72.8)	879 (82.1)	<0.001
Heterosexual	228 (13.0)	38 (19.6)	80 (16.3)	110 (10.3)	<0.001
Bisexual	88 (5.0)	0 (0.0)	24 (4.9)	64 (6.0)	0.002
IVDU	65 (3.7)	26 (13.4)	26 (5.3)	13 (1.2)	<0.001
Unknown	9 (0.5)	0 (0.0)	4 (0.8)	5 (0.5)	0.384
Comorbidities, *n* (%)					
Chronic kidney disease	7 (0.4)	0 (0.0)	2 (0.4)	5 (0.5)	0.637
Diabetes mellitus	41 (2.3)	7 (3.6)	16 (3.3)	18 (1.7)	0.074
Congestive heart failure	3 (0.2)	0 (0.0)	0 (0.0)	3 (0.3)	0.382
Solid tumour	13 (0.7)	1 (0.5)	4 (0.8)	8 (0.7)	0.919
Haematological cancer	5 (0.3)	2 (1.0)	2 (0.4)	1 (0.1)	0.066
Autoimmune disease	12 (0.7)	3 (1.5)	3 (0.6)	6 (0.6)	0.300
Organ transplantation	2 (0.1)	2 (1.0)	0 (0.0)	0 (0.0)	<0.001
Mean CD4+ count at presentation, cells/L (SD)	283 (214)	319 (229)	284 (227)	275 (204)	0.031
Subgroup of CD4+ count, cells/L (%) ^a^					
≤200	685 (39.0)	70 (36.1)	199 (40.4)	416 (38.8)	0.566
201–350	492 (28.0)	52 (26.8)	122 (24.8)	318 (29.7)	0.125
351–500	320 (18.2)	25 (12.9)	99 (20.1)	196 (18.3)	0.086
≥501	260 (14.8)	47 (24.2)	72 (14.6)	141 (13.2)	<0.001
HIV VL≥100,000 copies/mL, *n* (%) ^a^	698 (39.7)	86 (44.3)	205 (41.7)	407 (38.0)	0.148
HIV stage at presentation by 2014 CDC definition [48], *n* (%)					
Stage 0 (Acute HIV)	124 (7.1)	7 (3.6)	36 (7.3)	81 (7.6)	0.136
Stage 1 (CD4+ count ≥500 cells/L)	231 (13.1)	41 (21.1)	66 (13.4)	124 (11.6)	0.001
Stage 2 (CD4+ count 200–499 cells/L)	729 (41.5)	69 (35.6)	195 (39.6)	465 (43.7)	0.077
Stage 3 (AIDS)	673 (38.3)	77 (39.7)	195 (39.6)	401 (37.4)	0.649
AOIs during observation period					
Patients with AOI(s) at HIV presentation, *n* (%)	337 (19.2)	42 (21.6)	100 (20.3)	195 (18,2)	0.4
Patients who developed AOI(s) within the study period, *n* (%)	414 (23.6)	50 (25.8)	129 (26.2)	235 (21.9)	0.134

Abbreviations: AIDS, acquired immune deficiency syndrome; AOI, AIDS-related opportunistic illness; HIV, human immunodeficiency virus; IVDU, intravenous drug user; MSM, man who has sex with men; SD, standard deviation; VL, viral load. ^a^ Assayed within 6 months of enrolment.

**Table 2 jcm-08-00163-t002:** Spectrum of AIDS-related opportunistic illnesses (AOIs) with associated median CD4+ lymphocyte counts over the three study periods.

	Total		2001–2005	2006–2010	2011–2015	*p*
	No. of Episodes, *n* = 635	CD4^+^ T cell, cell/mm^3^ (Median, IQR)	No. of Episodes, *n* = 85	No. of Episodes, *n* = 207	No. of Episodes, *n* = 343	
Opportunistic infection, *n* (%)						
*Pneumocystis jirovecii* pneumonia	266 (57.8)	27 (51)	33 (58.9)	84 (57.1)	149 (58.0)	0.971
Cytomegalovirus disease (other than liver, spleen, or nodes)	55 (12.0)	25 (75)	3 (5.4)	14 (9.5)	38 (14.8)	0.078
*Mycobacterium tuberculosis* infection	78 (17.0)	87 (167)	22 (39.3)	32 (21.8)	24 (9.3)	<0.001
Wasting syndrome	47 (10.2)	50.5 (95)	6 (10.7)	12 (8.2)	29 (11.3)	0.604
Candidiasis (oesophagus, bronchi, trachea, lung)	54 (11.7)	22 (56)	13 (23.2)	17 (11.6)	24 (9.3)	0.014
Cryptococcosis, extrapulmonary	37 (8.0)	31 (45)	4 (7.1)	15 (10.2)	18 (7.0)	0.505
Disseminated *Mycobacterium avium* complex infection or *M. kansasii*	30 (6.5)	35 (56)	2 (3.6)	11 (7.5)	17 (6.6)	0.599
Cryptosporidiosis, chronic intestinal	6 (1.3)	28 (149)	0 (0.0)	1 (0.7)	5 (1.9)	0.367
HIV encephalopathy	7 (1.5)	121 (207)	0 (0.0)	3 (2.0)	4 (1.6)	0.568
HSV, chronic ulcer greater >1 month; or bronchitis, pneumonitis, or oesophagitis	5 (1.1)	66.5 (137)	0 (0.0)	0 (0.0)	5 (1.9)	0.136
Salmonellosis septicaemia, recurrent	5 (1.1)	32 (31)	0 (0.0)	1 (0.7)	4 (1.6)	0.505
Recurrent pneumonia	3 (0.7)	25	0 (0.0)	1 (0.7)	2 (0.8)	0.806
Progressive multifocal leukoencephalopathy	3 (0.7)	120	1 (1.8)	0 (0.0)	2 (0.8)	0.343
*Toxoplasma* encephalitis	2 (0.4)	45	1 (1.8)	0 (0.0)	1 (0.4)	0.221
*Talaromyces marneffei*, disseminated or extrapulmonary	2 (0.4)	18	0 (0.0)	1 (0.7)	1 (0.4)	0.794
Histoplasmosis, disseminated or extrapulmonary	0 (0.0)	N/A	0 (0.0)	0 (0.0)	0 (0.0)	N/A
Coccidioidomycosis, disseminated or extrapulmonary	0 (0.0)	N/A	0 (0.0)	0 (0.0)	0 (0.0)	N/A
Isosporiasis, chronic intestinal (>1 month)	0 (0.0)	N/A	0 (0.0)	0 (0.0)	0 (0.0)	N/A
Opportunistic malignancy						
Kaposi’s sarcoma	16 (3.5)	24 (207)	0 (0.0)	9 (6.1)	7 (2.7)	0.063
Lymphoma	18 (3.9)	109 (153)	0 (0.0)	5 (3.4)	13 (5.1)	0.194
Invasive cervical cancer	1 (0.2)	N/A	0 (0.0)	1 (0.7)	0 (0.0)	0.344

Abbreviations: HIV, human immunodeficiency virus; HSV, herpes simplex virus; IQR, interquartile range; N/A, not applicable.

**Table 3 jcm-08-00163-t003:** Logistic regression model for factors associated with MTB disease at presentation among patients newly diagnosed with HIV infection from 2001 to 2015.

	Univariable Analysis		Multivariable Analysis	
	Crude OR (95% CI)	*p*	Adjusted OR (95% CI)	*p*
Age, per 10-year increase	1.68 (1.38–2.04)	<0.001	1.40 (1.08–1.79)	0.009
Male	0.4 (0.14–1.15)	0.089	0.90 (0.27–3.01)	0.86
HIV transmission route				
Homosexual	Reference		Reference	
Heterosexual	2.50 (1.38–4.55)	0.003	1.01 (0.49–2.12)	0.97
Bisexual	0.77 (0.18–3.25)	0.72	0.62 (0.14–2.73)	0.53
IVDU	3.37 (1.38–8.27)	0.008	2.56 (0.92–7.12)	0.072
Unknown	0.00 (N/A)	0.99	0.00 (N/A)	0.99
Period				
Period 1 (2001–2005)	Reference		Reference	
Period 2 (2006–2010)	0.47 (0.25–0.86)	0.015	0.43 (0.22–0.83)	0.012
Period 3 (2011–2015)	0.16 (0.08–0.30)	<0.001	0.16 (0.08–0.32)	<0.001
Subgroup of CD4 cell count at presentation				
CD4 count ≥ 501 cells/µL	Reference		Reference	
CD4 count 351–500 cells/µL	0.54 (0.09–3.25)	0.50	0.78 (0.13–4.80)	0.79
CD4 count 201–350 cells/µL	1.06 (0.26–4.26)	0.94	1.43 (0.34–5.96)	0.62
CD4 count < 200 cells/µL	7.18 (2.22–23.20)	0.001	7.71 (2.20–27.05)	0.001
VL≥100,000 copies/mL	3.50 (2.04–5.98)	<0.001	1.60 (0.88–2.90)	0.12
Diabetes mellitus	0.66 (0.09–4.85)	0.68	0.23 (0.03–1.87)	0.17

Abbreviations: AIDS, acquired immune deficiency syndrome; HIV, human immunodeficiency virus; IVDU, intravenous drug user; MSM, man who has sex with men; N/A, not applicable; SD, standard deviation; VL, viral load.

**Table 4 jcm-08-00163-t004:** Cox regression hazard model for factors associated with MTB disease during follow-up among patients newly diagnosed with HIV infection from 2001 to 2015.

	Univariable Analysis		Multivariable Analysis	
	Crude HR (95% CI)	*p*	Adjusted HR (95% CI)	*p*
Age, per 10-year increase	1.12 (0.69–1.81)	0.66	0.96 (0.50–1.86)	0.91
Male	21.07 (0.00–9,912,364.81)	0.647	371,294.35 (0.00–∞)	0.99
HIV transmission route				
Homosexual	Reference		Reference	
Heterosexual	0.60 (0.08–4.72)	0.63	0.59 (7–4.91)	0.63
Bisexual	0.00 (0.00–∞)	0.98	0.00 (0.00–∞)	0.99
IVDU	9.29 (2.91–29.65)		5.63 (1.44–21.99)	0.013
Unknown	0.00 (N/A)	0.99	0.00 (N/A)	0.99
Period				
Period 1 (2001–2005)	Reference		Reference	
Period 2 (2006–2010)	1.27 (0.35–4.71)	0.72	1.08 (0.28–4.14)	0.91
Period 3 (2011–2015)	0.16 (0.03–0.78)	0.02	0.18 (0.04–0.91)	0.038
Subgroup of CD4 cell count at presentation				
CD4 count ≥ 501 cells/µL	Reference		Reference	
CD4 count 351–500 cells/µL	0.41 (0.07–2.22)	0.30	0.68 (0.12–3.88)	0.67
CD4 count 201–350 cells/µL	0.51 (0.13–2.03)	0.34	0.80 (0.19–3.29)	0.75
CD4 count < 200 cells/µL	0.52 (0.14–1.94)	0.33	1.01 (0.23–4.45)	0.99
VL≥100,000 copies/mL	1.09 (0.39–3.06)	0.87	1.20 (0.38–3.74)	0.76
Diabetes mellitus	0.05 (0.00–170,215.51)	0.693	0.00 (0.00–∞)	0.99

Abbreviations: AIDS, acquired immune deficiency syndrome; HIV, human immunodeficiency virus; IVDU, intravenous drug user; MSM, man who has sex with men; N/A, not applicable; SD, standard deviation; VL, viral load.

**Table 5 jcm-08-00163-t005:** Demographics, laboratory findings, and proportion of extrapulmonary tuberculosis (TB) among 78 patients with MTB disease.

	Total (2001–2015) *n* = 78	Period 1 (2001–2005) *n* = 22	Period 2 (2006–2010) *n* = 32	Period 3 (2011–2015) *n* = 24	*p*-Value
Age at event, *n* (%)	36.1 (10.2)	35.9 (10.7)	36.8 (10.3)	35.3 (10.2)	0.863
Male, *n* (%)	73 (93.6)	22 (100)	28 (87.5)	23 (95.8)	0.158
Event within 90 days of enrolment, *n* (%)	66 (84.6)	20 (90.9)	27 (84.4)	19 (79.2)	0.544
With extrapulmonary TB, *n* (%)	31 (39.7)	8 (36.4)	11 (34.4)	12 (50.0)	0.462
CD4+ counts <200 cells/L, *n* (%)	61 (78.2)	17 (77.3)	25 (78.1)	19 (79.2)	0.988
VL on event (>100,000 copies/mL), *n* (%)	47 (60.3)	13 (59.1)	20 (62.5)	14 (58.3)	0.943
Attributable mortality, *n* (%)	8 (10.3)	3 (13.6)	3 (9.4)	2 (8.3)	0.820

Abbreviations: TB, tuberculosis; VL, viral load.

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
