# Peer review of "Changing Spectrum of Opportunistic Illnesses among HIV-Infected Taiwanese Patients in Response to a 10-Year National Anti-TB Programme"

_jcm, 2019, doi:10.3390/jcm8020163_

Reviewer 1 Report

The authors of the reviewed manuscript have discussed the AIDS-related opportunistic illnesses (AOIs) among HIV-infected Taiwanese patients. This problem has been persistent and occurs in many developing and developed countries. That is why, it is so important to find the proper way/programme to fight with these diseases.

Data (including statistic) are presented in very clear way, diagrams and tables help to look overall at the patients’ diversity. I also like the authors use summary at the end of some sections to emphasize what is the most important.

There are minor mistakes/concerns such as (but not limited to):

-Please explain the definition of AIDS-related opportunistic illnesses (AOIs) and give some examples (Introduction section);

- Please explain what those abbreviations mean: KMUH and VGHKS (Materials and Methods section);

- Does anti-TB programme mean that the patients were treated only against TB? Besides the significant decrease of the incidence of AOIs with MTB disease the other incidences of AOIs declined too (candidiasis).

-Please explain what exactly mean: preprogramme phase, early programme phase and late programme phase, are they any other differences between those three periods besides the time, why did you divide this programme in this way? I am confused about that. If patients were enrolled in different periods of time why did you call them: “PREprogramme”, “EARLY programme” and “LATE programme”?

-Line 304: I believe it should be: ”esophageal candidiasis” instead of: “oesophageal candidiasis”

Author Response

Response to Reviewer 1 Comments

January 16, 2019

Emmanuel Andrès

Editor-in-Chief

Journal of Clinical Medicine

Dear Emmanuel Andrès

Re: Manuscript reference jcm-430481

Please find attached the revised version of our manuscript “Changing spectrum of opportunistic illnesses among HIV-infected Taiwanese patients in response to a 10-year national anti-TB programme” which we would like to resubmit for publication as an Article in Journal of Clinical Medicine, section Epidemiology & Public Health.

We found the comments of the reviewers highly insightful and they enabled us to significantly improve the quality of our manuscript. Please find our point-by-point response to each comment by the reviewer in the following pages.

Revisions in the text are shown using yellow highlight. We hope that the revisions in the manuscript and our accompanying responses will be sufficient to make our manuscript suitable for publication in Journal of Clinical Medicine, section Epidemiology & Public Health.

We look forward to hearing from you at your earliest convenience.

Sincerely,

Hung-Chin Tsai

Division of Infectious Diseases, Department of Medicine, Kaohsiung Veterans General Hospital, 386 Ta-Chung 1st Rd., Kaohsiung 813, Taiwan.

Tel: +886-7-342121 x1540

Fax: +886-7-3468292

E-mail: hctsai1011@yahoo.com.tw

Responses to the comments of Reviewer 1: 

Point 1

Please explain the definition of AIDS-related opportunistic illnesses (AOIs) and give some examples (Introduction section);

Response 1:

We appreciate the reviewer’s recommendation. We have revised the section in Introduction, as follows:

AOIs are infections or malignancy that occur more frequently and are more severe in host with weakened immune systems by HIV, such as Pneumocystis jirovecii pneumonia (PJP), toxoplasmosis of brain, and Kaposi’s sarcoma [12]. Page 4, line56-58.

Point 2

Please explain what those abbreviations mean: KMUH and VGHKS (Materials and Methods section);

Response 2:

We appreciate the reviewer’s observation, and have revised the section in Method, as follows:

This retrospective cohort study was conducted at Kaohsiung Medical University Hospital (KMUH) and Kaohsiung Veterans General Hospital (VGHKS), the two largest referral centres for patients with HIV in southern Taiwan. Page 7, line 107-109.

Point 3

Does anti-TB programme mean that the patients were treated only against TB? Besides the significant decrease of the incidence of AOIs with MTB disease the other incidences of AOIs declined too (candidiasis).

Response 3:

We appreciate your recommendation, and would like to include the explanation for this in the following lines.

As we have mentioned in the section of Introduction, anti-TB programme indicate “National Mobilization Plan to Halve TB in 10 Years” launched by Taiwan government to echo the Global Plan to Stop TB 2006–2015 advocated by the World Health Organization. The 10-year programme is a comprehensive, integrated plan to control, treat and prevent the disease burden of TB in Taiwan, without dealing with other AIDS-related opportunistic illness.

In order not to make confusion, we have revised the section in Introduction, as follows: TB has had a major public health impact on southeast Asian countries [31,32], including Taiwan [33,34]. In line with the Global Plan to Stop TB 2006–2015 advocated by the World Health Organization (WHO), Taiwan launched a programme aimed at halving TB incidence from 72.5 to 36 cases per 100,000 people by 2015, known as the National Mobilization Plan to Halve TB in 10 Years (Phase 1, 2006–2010 [35]; Phase 2, 2011–2015 [36]). The Phase 1 5-year programme was approved by Executive Yuan in 2006. With the implement of the Phase 1 programme, the number of new cases of MTB in Taiwan each year has declined from 16,472 in 2005 to 14,265 in 2008 [36]. After assessing the results of the Phase 1 programme, in coordination with The Global Plan to Stop TB 2006-2015 recommended by the Stop TB Partnership, the Department of Health approved the Phase 2 5-year programme to reduce TB burden in Taiwan. The 10-year plan aimed to develop a comprehensive national programme that strengthened resources at all levels of TB control and prevention, to intensify case-finding and reporting strategy, to improve the quality of diagnosis and therapy of tuberculosis, to implement case management, and to scale up the knowledge and awareness of the pubic to tuberculosis prevention and control. Page 4-5, line 68-83.

Point 4

Please explain what exactly mean: preprogramme phase, early programme phase and late programme phase, are they any other differences between those three periods besides the time, why did you divide this programme in this way? I am confused about that. If patients were enrolled in different periods of time why did you call them: “PREprogramme”, “EARLY programme” and “LATE programme”?

Response 4:

We appreciate your recommendation, and would like to include the explanation for this in the following lines.

The whole 10-year National Mobilization Plan to Halve TB in 10 Years can be divided into phase 1 and phase 2 programme. The Phase 1 5-year programme was first approved by Executive Yuan in 2006. After assessing the results of the Phase 1 programme, in coordination with The Global Plan to Stop TB 2006-2015 recommended by the Stop TB Partnership, the Department of Health approved the Phase 2 5-year programme in the continuing efforts to reduce TB burden in Taiwan. The aim of the present study is to evaluate the evolving AOI incidence rate and spectrum among newly diagnosed cases of HIV infection before programme, in the phase 1 programme, and in the phase 2 programme.

In order not to make confusion, we have revised the section in Introduction, as follows:

TB has had a major public health impact on southeast Asian countries [31,32], including Taiwan [33,34]. In line with the Global Plan to Stop TB 2006–2015 advocated by the World Health Organization (WHO), Taiwan launched a programme aimed at halving TB incidence from 72.5 to 36 cases per 100,000 people by 2015, known as the National Mobilization Plan to Halve TB in 10 Years (Phase 1, 2006–2010 [35]; Phase 2, 2011–2015 [36]). The Phase 1 5-year programme was approved by Executive Yuan in 2006. With the implement of the Phase 1 programme, the number of new cases of MTB in Taiwan each year has declined from 16,472 in 2005 to 14,265 in 2008 [36]. After assessing the results of the Phase 1 programme, in coordination with The Global Plan to Stop TB 2006-2015 recommended by the Stop TB Partnership, the Department of Health approved the Phase 2 5-year programme to reduce TB burden in Taiwan.

Page 4-5, line 68-78.

Therefore, the purpose of the present study was to estimate the evolving AOI incidence rate and spectrum among newly diagnosed cases of HIV infection at the two largest referral centres for patients with HIV in southern Taiwan in response to a 10-year national anti-TB programme. Patients were stratified into three periods: period 1 (Before programme, 2001–2005), period 2 (Phase 1 programme, 2006–2010), and period 3 (Phase 2 programme, 2011–2015). We further analysed factors associated with MTB disease at presentation and during follow-up. Page 6, line 98-104.”

Point 5

Line 304: I believe it should be: ”esophageal candidiasis” instead of: “oesophageal candidiasis”

Response 5:

We appreciate the reviewer’s recommendation. We have revised the problem.

Reviewer 2 Report

This manuscript reports the results of a study examining rates of AIDS-related opportunistic illnesses (AOIs) among patients newly diagnosed with HIV infection in Taiwan. The researchers aimed to gain a better understanding of the impact of a nationwide anti-tuberculosis program on AOIs. Additionally, the authors explored factors associated with Mycobacterium tuberculosis (MTB) at presentation and during follow-up.

The authors did a very good job overall, I offer some suggestions that mayimprove the quality of the paper.

Introduction: This is a very concise description of the background. It may be helpful to the reader if the authors clarify the “specific opportunistic illness” mentioned in line 61 and give some examples of different opportunistic infections that can occur in different regions.

Additionally, there are no clear research questions or hypotheses stated before the authors go into the Methodology section. The authors state that they analyze factors associated with MTB disease at presentation and during follow up. What particular factors are being examined?  The introduction could be strengthened by providing the reader with information on these factors and their known impact on MTB and other AOIs.

Methods:

The authors should explain what KMUH and VGHKS are, provide a brief description and explanation of why those sites were chosen.

The authors provided clear definitions of terms and criteria for diagnosing the different AOIs.

The authors provided a clear and thorough description of the statistical analysis plan.

Results/Discussion: The authors provided a very clear description of patient characteristics. The use of the figure is helpful for understanding how cases were selected for inclusion in the analysis and how they were categorized by time period.

The presentation of the results on general AOI incidence is very clear.

I have a few suggestions for the AOI spectrum results. I suggest changing the sentence regarding CMV disease starting on line 242 for clarification. 

Although not statistically significant, there was a notable increase in the incidence of CMV disease from 5.4% in period 1 to 14.8% in period 3 (p=0.078).

I suggest changing the last sentence of the paragraph (starting at line 244) so that it is clear that the significant changes across study periods were specific to the two AOIs (candidiasis and MTB). You indicated that the “most significant” changes were for tuberculosis and candidiasis when in fact they were the only significant changes.

In summary, the specific AOI incidence changed significantly across the three study periods for candidiasis and MTB only.

The presentation of factors associated with MTB at presentation and follow up is clear.

In your discussion section, some of the information on regional differences might have been better placed in your introduction. (See note above regarding suggested information to include in the introduction section). You could then briefly remind the reader of these regional differences in the discussion and discuss the differences in your findings compared to other studies done in the region of interest.

You stated that “observational data are vulnerable to inherent biases.” This statement seems out of place given the significant effort the authors used to describe the medical tests used to diagnose and the reliance on medical records as the source of the data. I am not sure these should be classified as observational data.

You also state that the results may not be generalizable to all of Taiwan. This is likely true, but explain why. For example, I noted above a lack of information on how the sites were chosen.  This information may explain the degree to which the results may or may not be generalizable.

The authors provided a good conclusion section including information on practical implications for the research.

Author Response

Response to Reviewer 2 Comments

January 16, 2019

Emmanuel Andrès

Editor-in-Chief

Journal of Clinical Medicine

Dear Emmanuel Andrès

Re: Manuscript reference jcm-430481

Please find attached the revised version of our manuscript “Changing spectrum of opportunistic illnesses among HIV-infected Taiwanese patients in response to a 10-year national anti-TB programme” which we would like to resubmit for publication as an Article in Journal of Clinical Medicine, section Epidemiology & Public Health.

We found the comments of the reviewers highly insightful and they enabled us to significantly improve the quality of our manuscript. Please find our point-by-point response to each comment by the reviewer in the following pages.

Revisions in the text are shown using yellow highlight. We hope that the revisions in the manuscript and our accompanying responses will be sufficient to make our manuscript suitable for publication in Journal of Clinical Medicine, section Epidemiology & Public Health.

We look forward to hearing from you at your earliest convenience.

Sincerely,

Hung-Chin Tsai

Division of Infectious Diseases, Department of Medicine, Kaohsiung Veterans General Hospital, 386 Ta-Chung 1st Rd., Kaohsiung 813, Taiwan.

Tel: +886-7-342121 x1540

Fax: +886-7-3468292

E-mail: hctsai1011@yahoo.com.tw

Responses to the comments of Reviewer 2:

Point 1

Introduction: This is a very concise description of the background. It may be helpful to the reader if the authors clarify the “specific opportunistic illness” mentioned in line 61 and give some examples of different opportunistic infections that can occur in different regions.

Response 1:

We appreciate the reviewer’s recommendation. We have revised the section in Introduction, as follows:

AOIs are infections or malignancy that occur more frequently and are more severe in host with weakened immune systems by HIV, such as Pneumocystis jirovecii pneumonia (PJP), toxoplasmosis of brain, and Kaposi’s sarcoma [12]. AOIs vary widely depending on the studied region (such as Mycobacterium tuberculosis [MTB] as the most common aetiologies in Philippines [13], India [14], and China [8]; PJP in Japan [13] and Australia [15]; esophageal candidiasis in Poland [16]; and cerebral toxoplasmosis in Lebanon [17]), the environmental prevalence of the microorganisms (such as MTB [18-21], Penicillium marneffei [8,22], Toxoplasma gondii [23], and Cryptococcus neoformans [24,25]), and the definitions of AOI [19,26-29]. Page 4, line 56-64.

Point 2

Additionally, there are no clear research questions or hypotheses stated before the authors go into the Methodology section. The authors state that they analyze factors associated with MTB disease at presentation and during follow up. What particular factors are being examined?  The introduction could be strengthened by providing the reader with information on these factors and their known impact on MTB and other AOIs.

Response 2:

We appreciate the reviewer’s recommendation. The scientific questions of the present study have been re-written in the 4th paragraph in Introduction section, as follows:

However, adequate data to investigate the evolving AOI incidence rate and spectrum among HIV-infected patients in Taiwan after the implementation of the 2006–2015 national TB programmes are lacking, particularly with regard to MTB disease [34-37]. Moreover, a study conducted in southern Africa revealed a possible interaction between MTB and other AOIs, such as cryptococcal infection and PJP, in HIV-infected patients [13], further reinforcing the need to update AOI surveillance in Taiwan following the 10-year national TB programme. In addition, previous studies conducted in Western Europe found advanced HIV infection, non-white ethnicity, migrants from high TB incidence, intravenous drug user (IVDU) as risk factors of MTB disease in HIV-infected patients [38-40]. However, these studies may not be applied to Taiwan, which have a higher TB incidence rate than Western Europe have [26]. Page 5, line 87-97.

Point 3

The authors should explain what KMUH and VGHKS are, provide a brief description and explanation of why those sites were chosen.

Response 3:

We have revised the section in Method, as follows:

This retrospective cohort study was conducted at Kaohsiung Medical University Hospital (KMUH) and Kaohsiung Veterans General Hospital (VGHKS), the two largest referral centres for patients with HIV in southern Taiwan. Page 7, line 107-109.

Point 4

I have a few suggestions for the AOI spectrum results. I suggest changing the sentence regarding CMV disease starting on line 242 for clarification.

Although not statistically significant, there was a notable increase in the incidence of CMV disease from 5.4% in period 1 to 14.8% in period 3 (p=0.078).

Response 4:

We appreciate the reviewer’s suggestion. We have revised the sentence as reviewer’s suggestion:

Although not statistically significant, there was a notable increase in the incidence of CMV disease from 5.4% in period 1 to 14.8% in period 3 (P=0.078). Page 18, line 261-263

Point 5

I suggest changing the last sentence of the paragraph (starting at line 244) so that it is clear that the significant changes across study periods were specific to the two AOIs (candidiasis and MTB). You indicated that the “most significant” changes were for tuberculosis and candidiasis when in fact they were the only significant changes.

In summary, the specific AOI incidence changed significantly across the three study periods for candidiasis and MTB only.

Response 5:

We appreciate the reviewer’s suggestion. We have revised the sentence as reviewer’s suggestion:

In summary, the specific AOI incidence changed significantly across the three study periods for candidiasis and MTB only. Page 18, line 263-265

Point 6:

In your discussion section, some of the information on regional differences might have been better placed in your introduction. (See note above regarding suggested information to include in the introduction section). You could then briefly remind the reader of these regional differences in the discussion and discuss the differences in your findings compared to other studies done in the region of interest.

Response 6:

We appreciate the reviewer’s recommendation. We have revised the section in Introduction, as follows:

AOIs vary widely depending on the studied region (such as Mycobacterium tuberculosis [MTB] as the most common aetiologies in Philippines [13], India [14], and China [8]; PJP in Japan [13] and Australia [15]; esophageal candidiasis in Poland [16]; and cerebral toxoplasmosis in Lebanon [17]), the environmental prevalence of the microorganisms (such as MTB [18-21], Penicillium marneffei [8,22], Toxoplasma gondii [23], and Cryptococcus neoformans [24,25]), and the definitions of AOI [19,26-29]. Page 4, line 58-64.

We have revised the section in Discussion, as follows:

Consistent with our findings, PJP was also the most common AOI in Japan [13]. However, regional differences in the AOI spectrum have been reported [8,13-17]. The observed regional differences in AOI distribution are likely attributable to a complex interaction between socioeconomic setting [50], geographical differences [8,51], genetic susceptibility [15], the implementation of highly active ART (HAART) [39], the prevalences of specific microorganisms [8,18-25], different definitions of AOIs [19,26-29], and different public health intervention methods, as also reported here.

Point 7:

You stated that “observational data are vulnerable to inherent biases.” This statement seems out of place given the significant effort the authors used to describe the medical tests used to diagnose and the reliance on medical records as the source of the data. I am not sure these should be classified as observational data.

Response 7:

We appreciate the reviewer’s recommendation. We have revised the section in Discussion, as follow:

First, some important information can’t be retrieved from retrospective chart review, such as contact history of TB index cases, whose behavioural and clinical characteristics have been reported to be associated with infectiousness of Mycobacterium tuberculosis [61], page 33, line 396-399.

Point 8

You also state that the results may not be generalizable to all of Taiwan. This is likely true, but explain why. For example, I noted above a lack of information on how the sites were chosen.  This information may explain the degree to which the results may or may not be generalizable.

Response 8:

We appreciate the reviewer’s recommendation. We have revised the section in Discussion, as follow:

Second, although the TB incidence rate have declined throughout Taiwan, southern Taiwan still have a higher TB incidence rate than northern Taiwan do [62,63]. Therefore, the results of the present study conducted in southern Taiwan may not be generalizable to all of Taiwan, page 33, line 399-402.
